

# Overexpression of IFITM3 predicts the high risk of lymphatic metastatic recurrence in pN0 esophageal squamous cell carcinoma after Ivor-Lewis esophagectomy

Yang Jia[1], Miao Zhang[2], Wenpeng Jiang[1], Zhiping Zhang[1], Shiting Huang[2] and Zhou Wang[1]

[1] Department of Thoracic Surgery, Shandong Provincial Hospital Affiliated to Shandong University, Jinan, Shandong, China
[2] Shandong Medical Imaging Research Institute, Shandong University, Jinan, Shandong, China

## ABSTRACT

**Background.** Recent studies have shown that the aberrant expression of IFITM3 is implicated in the lymph node metastasis of many malignancies. Our research aimed to investigate the expression of IFITM3 in pathological N0 (pN0) esophageal squamous cell carcinoma (ESCC) and its relationship with lymph node metastatic recurrence.

**Methods.** Immunohistochemistry (IHC) was used to examine the expression profile of IFITM3 in 104 pairs of samples. Each pair consisted of ESCC tissue and its adjacent normal mucosa (ANM). This aberrant expression was verified by reverse transcription-polymerase chain reaction (RT-PCR) with 20 tumor specimens with strong immunostaining and their mucosal tissues. In addition, 20 samples of low expression tissues and their ANMs were evaluated. Moreover, the correlations between the IFITM3 expression level and the clinicopathological variables, recurrence risk and overall survival (OS) of patients were analyzed.

**Results.** Both IHC and RT-PCR demonstrated that the IFITM3 expression level was significantly higher in tumor tissue than in ANM. Statistical analysis showed a significant correlation of IFITM3 expression with the T status of esophageal cancer ($p = 0.015$). In addition, IFITM3 overexpression was demonstrated to be not only an important risk factor of lymphatic metastatic recurrence but a significant prognostic factor in pN0 ESCC ($p < 0.005$).

**Conclusions.** Even pN0 ESCC patients will still experience lymphatic metastatic recurrence. The IFITM3 gene could be a predictor of lymphatic metastatic recurrence in pN0 ESCC after Ivor-Lewis esophagectomy.

Corresponding author
Zhou Wang,
wz620226@hotmail.com

## INTRODUCTION

Esophageal carcinoma (EC) is the sixth leading cause of mortality among various malignant tumors worldwide. Esophageal adenocarcinoma and esophageal squamous cell carcinoma (ESCC) are the two most common histological types. Both have an obvious characteristic geographic distribution, and China is a country with high incidence of ESCC (*Ferlay et al., 2010*). Despite the improvement in the diagnostic level and utilization of combined treatment modalities in recent years, the prognosis of ESCC patients remains poor. Even in pN0 ESCC, the 5-year survival rate is only approximately 70% after complete resection, and lymphatic metastatic recurrence is the main reason for the failure of the operation (*Eloubeidi et al., 2002*; *Visbal et al., 2001*). Therefore, to improve the long-term survival of ESCC patients, it is of great clinical significance to control locoregional lymph node metastatic recurrence after surgery.

Clinically, the number of metastasis-positive lymph nodes is usually used to evaluate the risk of lymphatic metastatic recurrence in locally advanced disease (*Law & Wong, 2001*). However, no reliable index has been used to predict this rate in pN0 ESCC. Although some molecules were previously used to stratify the recurrence risk in pN0 ESCC (*Li, Wang & Liu, 2009*; *Song et al., 2012*), none has been proven to be universally accepted and commonly used. Therefore, sensitive biological markers that may predict this recurrence risk are urgently needed.

Interferon-induced transmembrane protein 3 (IFITM3), also known as 1-8U, is one of the important members of the IFN-inducible transmembrane protein family. IFITM3 likely exerts profound influences on cell proliferation, migration, and invasion through the modulation of the Wnt/$\beta$-catenin signaling pathway and is implicated in the G0/G1 checkpoint to control the cell cycle of tumors (*Hu et al., 2014*; *Yang et al., 2013*; *Zhao et al., 2013*). It is overexpressed in many human malignancies, such as gastric cancer, colorectal tumor, breast cancer, glioma, and oral squamous cell carcinoma.

Previous research has shown that IFITM3 is upregulated in gastric cancer, which is correlated with tumor invasion and metastasis (*Hu et al., 2014*). Moreover, it has also been demonstrated to have a close relationship with the prognosis of colon cancer and was confirmed to be an independent risk factor for disease-free interval (*Li et al., 2011*). However, the clinicopathological significance and prognostic value of IFITM3 in ESCC patients remain unknown.

In this study, we sought to validate the relationships between the expression level of IFITM3 and the clinicopathological characteristics, lymphatic metastatic recurrence risk and overall survival of ESCC patients who had undergone Ivor-Lewis esophagectomy. We aimed to explore whether the IFITM3 gene can predict lymphatic metastatic recurrence in pN0 ESCC.

## MATERIALS AND METHODS

### Patients

From January 2008 to January 2010, patients with midthoracic ESCC who underwent Ivor-Lewis esophagectomy with two-field lymphadenectomy in our department (Provincial

**Table 1 Correlation of IFITM3 expression with clinicopathological characteristics of pN0 ESCC patients.**

| Variables | No. of patients | IFITM3 expression | | p value[a] | 3-year recurrence rate (%) | p value[b] |
|---|---|---|---|---|---|---|
| | | High 59 | Low 45 | | | |
| Age (years) | | | | 0.335 | | 0.550 |
| ≥50 | 83 | 45 | 38 | | 42.2 | |
| <50 | 21 | 14 | 7 | | 33.3 | |
| Gender | | | | 0.365 | | 0.889 |
| Male | 78 | 42 | 36 | | 41.0 | |
| Female | 26 | 17 | 9 | | 38.5 | |
| Tumor size (cm) | | | | 0.418 | | 0.193 |
| ≥5 | 40 | 25 | 15 | | 47.5 | |
| <5 | 64 | 34 | 30 | | 35.9 | |
| T status | | | | 0.015 | | **0.008** |
| T1 + T2 | 41 | 30 | 11 | | 24.4 | |
| T3 | 55 | 24 | 31 | | 49.1 | |
| T4a | 8 | 5 | 3 | | 62.5 | |
| Differentiation degree | | | | 0.249 | | 0.111 |
| Low | 25 | 17 | 8 | | 52.0 | |
| Moderate-high | 79 | 42 | 37 | | 36.7 | |
| IFITM3 overexpression | | | | | | **0.010** |
| Yes | | 59 | | | 50.8 | |
| No | | | 45 | | 26.7 | |

**Notes.**
[a] $\chi^2$ test.
[b] Log-rank test.

Hospital Affiliated to Shandong University, China) were eligible for this study. In total, this study enrolled 104 patients, including 83 men and 21 women, with ages ranging from 40 to 75 years (Clinicopathological data are listed in Table 1).

All patients met the following inclusion criteria: (1) According to 2009 Union for International Cancer Control (UICC) standard for midthoracic ESCC, Ivor-Lewis esophagectomy with two-field lymph node dissection was conducted to achieve complete resection (R0), and the proximal and distal incisal margins as well as lateral margin were pathologically examined without residual foci (*Arai et al., 2012*). At the same time, the average number of lymph nodes dissected was 18 ± 5.8 (ranging from 12 to 25); (2) Patients enrolled in the study were restaged after surgery according to TNM staging for esophageal cancer; (3) Without history of previous malignancies or other severe diseases that may influence the outcome of our follow-up; (4) Patients were not eligible if preoperative neoadjuvant chemotherapy or postoperative adjuvant treatment was administered.

## Surgical procedure of Ivor-Lewis esophagectomy

Four thoracic surgeons worked together to perform this type of surgery, and the thoracic operation was performed by two surgeons. The patient was placed in the 40°–45° left lateral decubitus position. After a right anterolateral thoracotomy, the chest was entered through the fourth intercostal space. The azygos vein arch was divided, and the esophagus was dissected from the esophagogastric junction to the apex of the chest. When the tumor invasion obviously extended outside the esophagus, the thoracic duct was routinely ligated above the diaphragm.

At the same time, an upper midline abdominal incision was made by another two surgeons, and the abdomen was explored. During mobilization of the stomach, care was taken to preserve the right gastroepiploic vessels and arcades. The left gastric artery and vein were isolated and doubly ligated at their origin. Pyloroplasty was not routinely performed. Then, the hiatus was enlarged and the stomach was pulled into the chest. An endto-side esophagogastric anastomosis was performed within the apex of the chest, and the stomach was secured in the mediastinum (*Chen et al., 2009a*; *Chen et al., 2009b*).

## Specimens

The ESCC tissue and ANM (more than 5 cm from the margin of ESCC) were collected from surgical specimens of each selected patient. At the same time, the ANM did not exhibit tumor infiltration, deterioration or necrosis upon macroscopic and microscopic examination.

This study was approved by the Ethics Committee of Provincial Hospital Affiliated to Shandong University, and the approval number was 2008081. Written informed consent was obtained from all the participants.

## Immunohistochemistry

The streptavidin-peroxidase immunohistochemical method was used to examine the expression of the IFITM3 protein. Formalin-fixed and paraffin-embedded surgical specimens were sequentially cut into 4-μm sections. Then, the sections were dewaxed, antigen retrieval and hydrogen peroxide incubation. Rabbit anti-IFITM3 monoclonal antibodies (GeneTex, Irvine, California, USA) were used at a dilution of 1:200 and incubated at 4 °C overnight. The monoclonal primary antibody was replaced by phosphate-buffered saline (PBS) as a negative control. Further experimental steps were followed according to the instructions of a secondary biotinylated antibody kit purchased from ZSGB Biotech (Beijing, China).

The expression of the IFITM3 protein was determined according to Sakakura's criteria. Two pathologists blinded to the clinical data were invited to evaluate the IHC sections independently. The outcome was calculated by combining the proportion with the staining intensity. The proportion was scored as follows: 0 (0–10%), 1 (11–25%), 2 (26–50%), 3 (51–75%), and 4 (75–100%). The staining intensity was scored: 0 (negative), 1 (weak), 2 (moderate), and 3 (strong). The final immunohistochemical score (IHS) was defined as the proportion score × staining intensity score. In this study, IHS ≥ 8 was considered to represent overexpression.

## RNA extraction and RT-PCR

Total RNA was extracted from fresh frozen tissue using Trizol (Invitrogen) according to the manufacturer's protocols. The purity of RNA was measured by UV spectrophotometer (NanoDrop 2000) and the OD 260/280 value ranging from 1.8 to 2.0 was used to reverse transcription. The detailed RT-PCR procedure was followed by a CWBio two-step RT-PCR kit (JiangSu, China). The primer sequences of the IFITM3 gene were 5′-CAAGGAGGAGCACGAGG-3′ (forward primer) and 5′-TTGAACAGGGACCAGACG-3′ (reverse primer). $\beta$-actin was used as internal control, and the primer sequences of $\beta$-actin were 5′-AGAGCCTCGCCTTTGCCGATCC-3′ (forward primer) and 5′-ATACACCCGCTGCTCCGGGTC-3′ (reverse primer). The PCR products of IFITM3 were further separated on 1% agarose gel electrophoresis. Azure C2000 (Azure Biosystems, Dublin, California, USA) was used for electrophoresis gelatin image formation analysis.

## Follow-up

According to our plan, patients were examined every 3–6 months after surgery, and the checklist was described in our previous study (*Akhtar et al., 2014*). We compared the imaging data preoperatively and postoperatively in detail to differentiate whether recurrence occurred or not. If the lymph nodes were swollen or the minor axis was more than 1 cm, the clinical diagnosis of lymphatic recurrence was made. Fine-needle aspiration biospy and PET-CT were also assisted to make the diagnosis of recurrence. The follow-up was ended in July 2013, and the complete 3-year follow-up data was reviewed.

## Statistical analysis

The $\chi^2$ test was used to analyze the relationship between IFITM3 expression and clinicopathological variables. The recurrence curves and survival curves were calculated by the Kaplan–Meier method. Univariate log-rank test and Cox regression analysis were respectively performed to evaluate the recurrence risk and prognostic factors. A satatistically significant difference was represented by a two-tailed *p* value less than 0.05. All statistical analyses were performed using SPSS version 17.0 (Chicago, Illinois, USA).

# RESULTS

## IFITM3 expression analysis in ESCC tissue and ANM

The immunohistochemistry assay was used to detect the expression level of the IFITM3 protein. Overexpression was presented as yellow or brownish yellow staining in the cytoplasm of the tumor cell. As is shown in Fig. 1C, the significant immunoreaction of positive expression can be readily differentiated. However, there was low or undetected staining in ANM (Fig. 1A). Furthermore, according to the criteria of IHS, we divided all the specimens into two groups: 59 cases (56.7%) were categorized as the overexpression group (Fig. 1C) and 45 cases (43.3%) were in the low expression group (Fig. 1B).

To verify this aberrant upregulation of IFITM3, we examined the mRNA expression level by RT-PCR with 20 pairs of specimens randomly selected from the overexpression group and 20 pairs of tissues that originated from the low expression group. The results

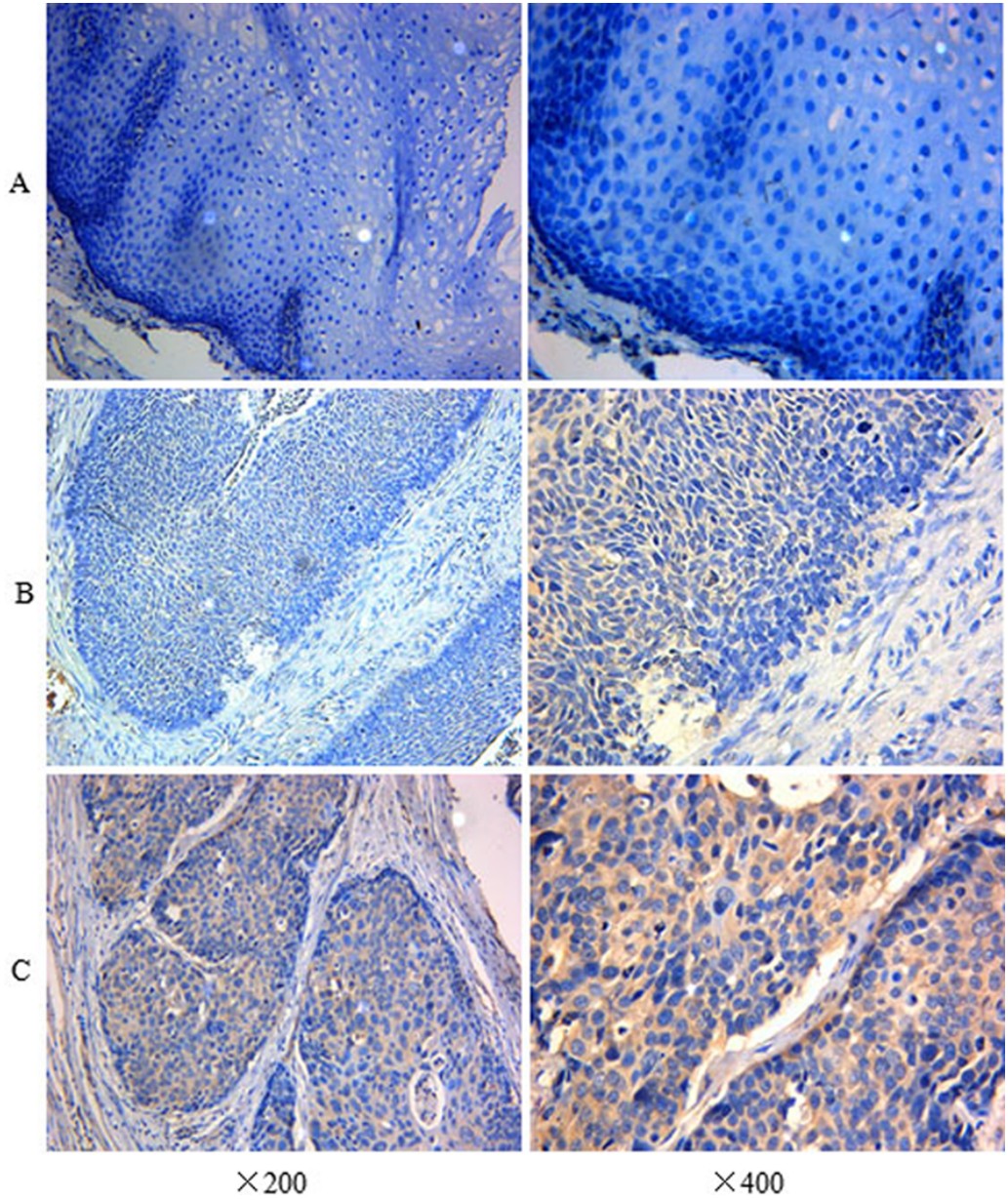

×200          ×400

**Figure 1  Immunohistochemistry assay of IFITM3 in ESCC tissue and ANM.** (A) Negative expression of IFITM3 in ANM (×200, ×400). (B) Low expression in ESCC tissue (×200, ×400). (C) Strong positive immunocreation of IFITM3 in the cytoplasm of ESCC tissue (×200, ×400).

showed that the mRNA expression level was consistent with protein expression as demonstrated by IHC (Fig. 2).

## IFITM3 expression and clinicopathological characteristics

According to the inclusion criteria mentioned above, a total of 104 ESCC patients were enrolled in this study with different ages, genders, tumor sizes, degrees of differentiation, T status and IFITM3 expression levels (Table 1). $\chi^2$ analysis demonstrated that the

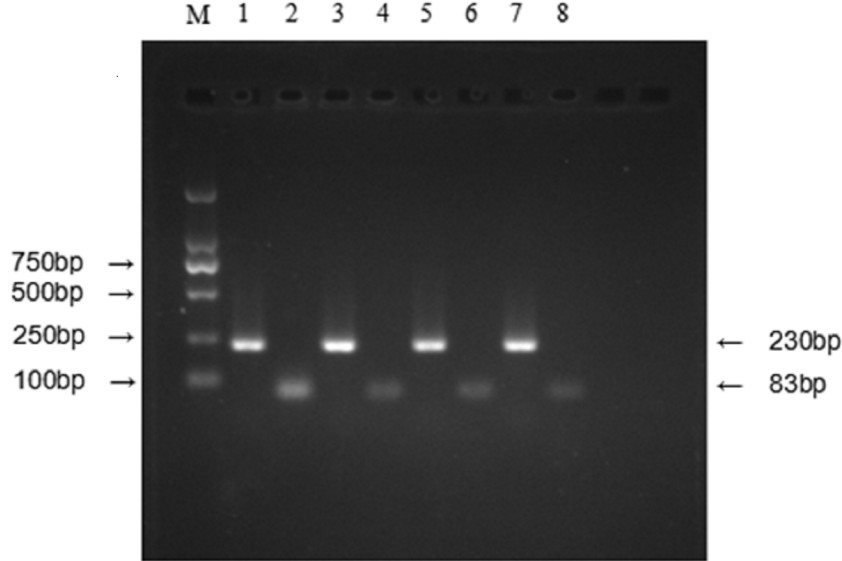

**Figure 2 Relative expression level of IFITM3 mRNA was detected by RT-PCR.** (M) Molecular marker. (1) and (2) respectively represent the mRNA expression level of $\beta$-actin and IFITM3 in tumor tissues with IFITM3 protein overexpression; (3) and (4) represent the mRNA expression level of $\beta$-actin and IFITM3 in their ANMs. (5) and (6) respectively represent the mRNA expression level of $\beta$-actin and IFITM3 in low IFITM3 protein expressed tumor tissues; (7) and (8) represent this expression level of $\beta$-actin and IFITM3 in their ANMs.

expression level of IFITM3 had a close relationship with T status of tumor ($p = 0.015$). In contrast, there were no significant differences between expression level and age, gender, tumor size and degree of differentiation ($p > 0.05$).

## IFITM3 expression and lymphatic metastatic recurrence

Through thorough follow-up, a total of 42 cases (40.4%) were confirmed to have first lymph node metastatic recurrence within 3 years, in which IFITM3 overexpression was detected in 30 patients (71.4%). In the low IFITM3 expression group, the 3-year lymphatic recurrence rate was only 26.7%. Conversely, in the overexpression group, this rate reached up to 50.8% (Table 1). As is shown in Fig. 3, Kaplan–Meier analysis showed that the recurrence rate was significantly increased in patients with IFITM3 overexpression, and the log-rank test calculated that these two curves were significantly different ($p = 0.010$).

In addition to the expression level of IFITM3, T status of tumor ($p = 0.008$) was also elucidated to be associated with the lymphatic recurrence in pN0 ESCC (Table 1). Multivariate Cox regression analysis revealed that these two variables were independent recurrence risk factors ($p < 0.05$, Table 2). Patients with early T status and low expression of IFITM3 may have a lower recurrence risk of lymphatic metastasis (Fig. 3).

## IFITM3 expression and overall survival

As is shown in our data, the 3-year OS rate of patients with IFITM3 protein overexpression and low expression were respectively 64.4% and 88.9%. Figure 4 demonstrated that the long survival of patients was associated with moderate-high differentiation, early T

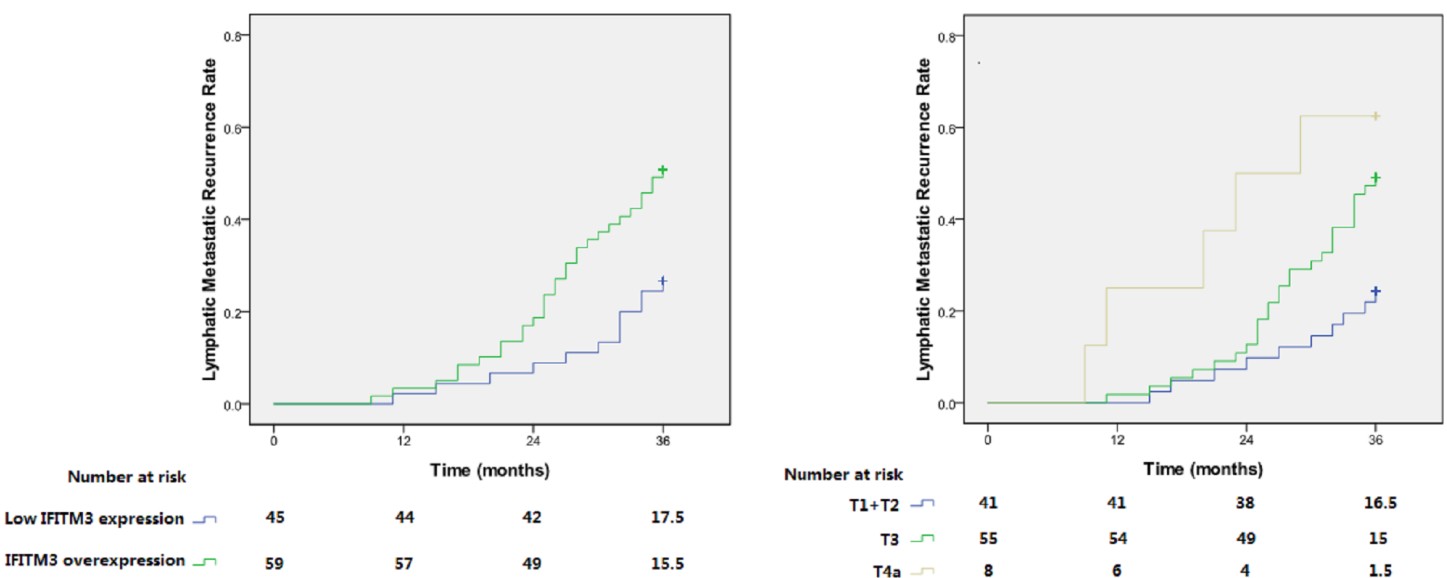

**Figure 3 Lymphatic metastatic recurrence curves for patients with different IFITM3 expression level and T status.** (A) and (B) respectively represent patients with IFITM3 overexpression ($p = 0.010$) and advanced T status ($p = 0.004$).

**Table 2 Multivariate Cox regression analysis of risk factors in pN0 ESCC.**

| | *B* | SE | Wald | *p* value | HR | 95%CI | |
|---|---|---|---|---|---|---|---|
| | | | | | | Lower | Upper |
| Age | 0.040 | 0.453 | 0.008 | 0.930 | 1.041 | 0.429 | 2.526 |
| Gender | 0.041 | 0.389 | 0.011 | 0.915 | 1.042 | 0.487 | 2.232 |
| Tumor size | 0.226 | 0.324 | 0.489 | 0.484 | 1.254 | 0.665 | 2.365 |
| T status | 0.904 | 0.253 | 12.783 | **0.000** | 2.470 | 1.505 | 4.054 |
| Differentiation | 0.311 | 0.350 | 0.790 | 0.374 | 1.365 | 0.687 | 2.712 |
| IFITM3 overexpression | 1.040 | 0.360 | 8.357 | **0.004** | 2.828 | 1.398 | 5.723 |

**Notes.**
B, regression coefficient; SE, standard error; Wald, Wald value; HR, hazard ratio; CI, confidence interval.

status of tumor and low IFITM3 expression. Univariate analysis and multivariate analysis revealed that these are independent and significant prognostic factors ($p < 0.05$, Table 3).

## DISCUSSION

ESCC is one of the most common neoplasms in China, with a high incidence of lymph node metastatic recurrence, especially in the mediastinum, neck and abdominal cavity (*Chen et al., 2007*). Even in pN0 ESCC, more than 40% of individuals exhibit micro-metastasis (*Wang et al., 2004*). Surgery is still considered to be the first-line treatment modality for ESCC patients with resectable lesions (*Hulscher et al., 2002*; *Olsen et al., 2011*), but the overall survival is not ideal and nearly half of patients will still experience tumor relapse (*Eloubeidi et al., 2002*; *Korst et al., 1998*; *Rice et al., 2001*; *Visbal et al., 2001*).

To date, there is no general treatment standard in China for ESCC; the NCCN esophageal cancer guidelines are often referenced in clinical practice. These guidelines sug-

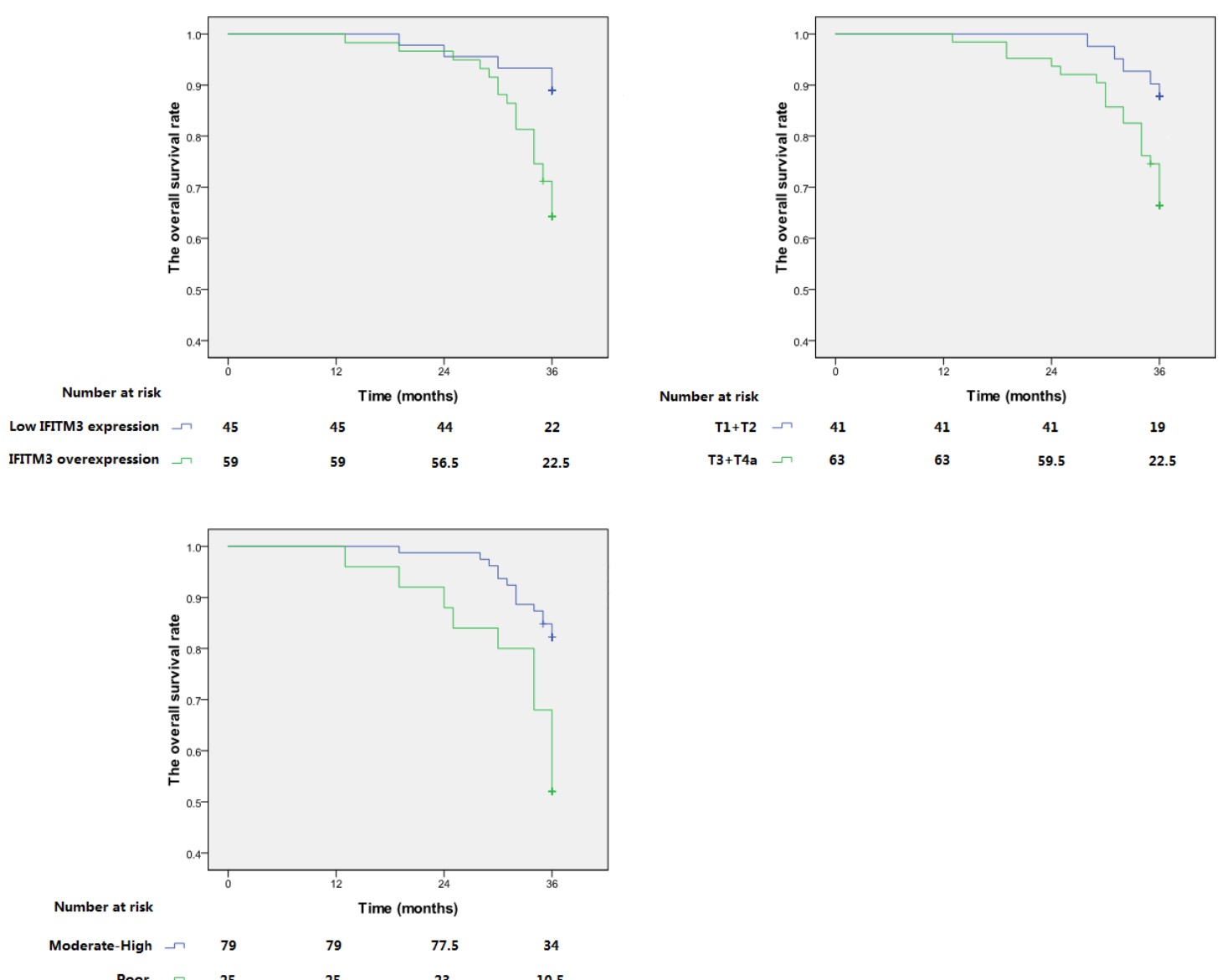

**Figure 4 Kaplan–Meier analysis of overall survival of ESCC patients.** Overall survival of patients with pN0 ESCC according to (A) IFITM3 expression level; (B) T status of tumor; (C) differentiation degree.

gest that patients should not receive adjuvant therapy after complete tumor resection, but individuals with advanced T status (above T2) should receive neoadjuvant chemotherapy before surgery. However, in China, patients tend to receive primary surgery if tumors can be completely resected and Ivor-Lewis esophagectomy via a thoracoabdominal two-field lymph node dissection is the main surgical modality. Compared with three-field lymph node dissection, the advantage of Ivor-Lewis esophagectomy is that the latent surgical trauma and complications can be effectively controlled; however, cervical lymph node dissection cannot be accomplished simultaneously. Therefore, for patients with a high risk

**Table 3** Univariate and multivariate analyses of overall survival for 104 cases of pN0 ESCC patients.

| Variables | Univariate analysis | | | Multivariate analysis | | |
|---|---|---|---|---|---|---|
| | HR | 95% CI | $p$ | HR | 95% CI | $p$ |
| Gender | | | | | | |
| Male versus female | 1.095 | 0.440–2.727 | 0.846 | – | | |
| Age (years) | | | | | | |
| ≥50 versus <50 | 1.388 | 0.478–4.028 | 0.546 | – | | |
| Tumor size (cm) | | | | | | |
| ≥5 versus <5 | 2.076 | 0.960–4.490 | 0.063 | – | | |
| T status | | | | | | |
| T1 + T2 versus T3 + T4a | 3.082 | 1.162–8.175 | **0.024** | 3.979 | 1.443–10.974 | **0.008** |
| Differentiation degree | | | | | | |
| Moderate-high versus low | 3.125 | 1.445–6.758 | **0.004** | 2.346 | 1.056–5.212 | **0.036** |
| IFITM3 overexpression | | | | | | |
| Yes versus no | 3.644 | 1.373–9.673 | **0.009** | 4.260 | 1.548–11.723 | **0.005** |

**Notes.**

Statistical analysis was performed using the proportional hazard model (Cox). Data considered significant ($p < 0.05$) in the univariate analysis were examined in the multivariate analysis.

HR, hazard ratio; CI, confidence interval.

$p$ values < 0.05 in bold font were considered significant.

of lymphatic metastasis, we believe that postoperative adjuvant therapy may act as a compensatory mechanism to control lymphatic recurrence after Ivor-Lewis esophagectomy.

In this study, we first found the differential expression of IFITM3 in tumor tissues and their ANMs, as well as the important clinicopathological significance of IFITM3. Our results were consistent with previous researches that demonstrated that IFITM3 is overexpressed in many hunman malignancies, such as gastric cancer, colorectal cancer, oral squamous cell carcinoma, glioma, and breast cancer. These findings suggested that IFITM3 may play important roles and maybe a molecular marker in pN0 ESCC.

Regarding to the prognostic value of IFITM3, previous studies have elicited contradictory conclusions for different cancers, which reflects the complexity of IFITM3 in different tumor microenvironments. For gastric cancer, *Hu et al. (2014)* suggested that IFITM3 overexpression is correlated with lymph node metastasis. *Li et al. (2011)* demonstrated that it was an important independent prognostic factor for disease-free interval and is upregulated in the nodal metastasis of colon tumors. Conversely, *Yang et al. (2013)* did not find an association between IFITM3 expression and lymph node metastasis in breast cancer. *El-Tanani et al. (2010)* even drew the opposite conclusion; they deemed that IFITM3 may inhibit the proliferation, development and metastasis of cancer by reducing the expression of osteopontin. However, to our knowledge, no study has demonstrated the prognostic significance of IFITM3 in ESCC. This question led us to explore whether IFITM3 could be a biomarker to evaluate the risk of lymph node metastatic recurrence in ESCC. The data from this study showed that the high incidence of lymphatic metastatic recurrence in pN0 ESCC was associated with advanced T status and IFITM3 overexpression. At the same time, the expression level of IFITM3 was also

elucidated to be an important prognostic factor. The findings strongly suggested that IFITM3 could serve as a biomarker to stratify the risk of recurrence and survival, and then play an important role in the selection of a treatment modality in pN0 ESCC.

A total of 104 patients with midthoracic ESCC in this study underwent Ivor-Lewis esophagectomy with two-field lymph node dissection. All of these patients had undergone theoretic R0 resection and had pathologically confirmed pN0 after surgery. However, during the follow-up period, 40.4% patients showed the first lymphatic metastatic recurrence within 3 years. For patients with a high risk of lymphatic metastatic recurrence, we believe that these findings have important clinical significance in choosing to accept adjuvant therapy to control lymph node recurrence. Previous studies have demonstrated that postoperative adjuvant radiotherapy can significantly reduce the lymphatic metastatic recurrence in ESCC. Combined with findings in this study and our previous studies, we believe that it is indispensible for pN0 ESCC patients with IFITM3 overexpression to receive postoperative adjuvant radiotherapy to control the metastatic recurrence of locoregional lymph nodes and in turn improve the survival.

However, this study was retrospective and had a limited sample size. Although this is the first time that IFITM3 was demonstrated to be a predictor for lymph node metastatic recurrence of ESCC patients, replication studies with different parameters and prospective and multicenter randomized studies are also needed to verify this prognostic significance.

## CONCLUSIONS

Our study demonstrated that IFITM3 expression has a close relationship with lymphatic metastatic recurrence. This could serve as an important biomarker to predict the lymph node metastatic recurrence in pN0 ESCC after Ivor-Lewis esophagectomy.

### Funding

Funding was provided by the Science and Technology Development Plan Project of Shandong Province (2014GSF118167). The funders had no role in study design, data collection and analysis, decision to publish, or preparation of the manuscript.

### Grant Disclosures

The following grant information was disclosed by the authors:
Science and Technology Development Plan Project of Shandong Province: 2014GSF118167.

### Competing Interests

The authors declare there are no competing interests.

### Author Contributions

- Yang Jia conceived and designed the experiments, performed the experiments, analyzed the data, contributed reagents/materials/analysis tools, wrote the paper, prepared figures and/or tables.

- Miao Zhang performed the experiments, analyzed the data, contributed reagents/materials/analysis tools, wrote the paper, prepared figures and/or tables.
- Wenpeng Jiang performed the experiments, contributed reagents/materials/analysis tools, reviewed drafts of the paper.
- Zhiping Zhang and Zhou Wang conceived and designed the experiments.
- Shiting Huang analyzed the data, contributed reagents/materials/analysis tools, reviewed drafts of the paper.

## Human Ethics

The following information was supplied relating to ethical approvals (i.e., approving body and any reference numbers):

This study was approved by the Ethic Committee of Provincial Hospital Affiliated to Shandong University. The approval number is 2008081.

## Supplemental Information

Supplemental information for this article can be found online at http://dx.doi.org/10.7717/peerj.1355#supplemental-information.

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
