# Peer review of "Overexpression of IFITM3 predicts the high risk of lymphatic metastatic recurrence in pN0 esophageal squamous cell carcinoma after Ivor-Lewis esophagectomy"

_PeerJ, doi:10.7717/peerj.1355_

## Round 0.1 · original submission · Major Revisions

The reviewers felt that your manuscript has value. However, there are a number of points that require clarifcation or more detail.Please address the following:
1. Edit the manuscript for readability and clarity using someone with good written English skills. See reviewer's comment particularly for the results section.
2. Discuss the 40% recurrence rate and why it is so high in this group. Is this because the patients were not getting standard of care chemotherapy or radiochemotherapy for more advanced tumors. What is the typical recurrence rate in the literature?
3. Provide additional details on the surgical technique including the number of surgeons.
4. In the Discussion comment on what might be recommended in a pN0 patient if the IFITM3 levels were high.
5. Consider deletion of the paragraph on McKeown esophagectomy as this is not the focus of this study.
6. Provide additional details on the number of lymph nodes that were dissected. In addition to the average, a range could be provided. The number of lymph nodes that were dissected could be included in Table 1.
7. Clarify figure 2. (See details from reviewer 2).
8. The authors should evaluate IFITM3 mRNA levels in a subset of additional tumors that showed low or medium expression by IFITM3 to confirm the correlation between protein and mRNA expression.
9. Clarify the comments from reviewer 2 on lymph node recurrence and how this was measured.
10. If possible include whether there is a correlation in IFITM3 levels with survival time (3 years/5 years).

Reviewer 1 ·

Basic reporting

The manuscript appears to adhere to all PeerJ policies and is written with standard formatting and structure. Figures are well done. Overall, the manuscript is well written with a few typographical errors (abstract: methods; results line 155)

Experimental design

The authors sought to determine whether IFITM3 serves as a predictor of regional recurrence in pN0 patients undergoing surgical treatment of ESCC. They analyzed 104 patients between 2008 to 2010 that were treated with Ivor-Lewis esophagectomy and two-field lymph node dissection. This study included only pN0 patients to identify predictors of regional recurrence. Patients did not receive adjuvant treatment.

IFITM3 expression was determined by IHC and RT-PCR in 20 specimens for confirmation.

Validity of the findings

Authors identified overexpression of IFITM3 in 56.7% of cases and that this expression was statistically related to T stage. 40.4% of patients developed regional recurrence within 3 years, and of these that recurred, 30/42 (71%) had IFITM3 overexpression, whereas in low IFITM3 expression the regional recurrence was 26.7%. The authors' statistical analysis showed that IFITM3 was thus a significant predictor of regional recurrence in this cohort. Multivariate analysis showed that IFITM3 was an independent predictor and not just a surrogate marker (such as for advanced T stage).

Additional comments

This is a very well designed study and well written manuscript. I have a few comments:

1. Please comment on 40.4% regional recurrence rate which seems high in a cohort treated with surgery alone. What are potential future directions - more extensive lymphadenectomy and/or adjuvant treatment?

2. Regional recurrence is after lymphadenectomy is related to surgical technique and thoroughness of the dissection - how many surgeons were involved and were similar lymphadenectomy techniques employed?

3. Related to #1, should IFITM3 overexpression warrant intensification of treatment in the form of adjuvant radiation treatment?

·

Basic reporting

A native speaker should revise the manuscript. Especially the results section is difficult to read.

In the discussion section the authors discuss suddenly McKeown esophagectomy. However in their study they only use Ivor-Lewis procedures. In the same paragraph the correctly write that adjuvant therapy is useful to improve prognosis. However their patients are not treated neither by neoadjuvant nor adjuvant therapy. The whole paragraph can be deleted, as it does not discuss any of their own findings.

Experimental design

The authors write that average lymph node dissection was greater than 12. Does that mean there were also patients with less lymph nodes dissected?

It is unclear why patients were not treated with chemotherapy or radiochemotherapy when having more advanced tumors. This is the current standard and if this study aims to allow conclusions for future treatment the patients should be treated according to these standards.

Validity of the findings

Table 1 should include the number of dissected lymph nodes with the median (+ min and max).

It is totally unclear to the reviewer what the authors want to show in Figure 2. The authors writer that in lanes 1 and 2 b-actin is shown. However there are bands with a different size. How can this be if they should show the same? The same is true for lanes 3 and 4.

Why did the authors only reconfirm IFITM3 mRNA expression in tumor tissue with high expression in IHC? The authors should do the same analyses in patients with low IHC expression and quantify the mRNA expression within the tumor samples. Otherwise the mRNA findings are not of any additional information.

The authors highlight that IFITM3 was differentially expressed according to the T stage, however they do not describe how and they omit to mention that low expression of IFITM3 correlated with higher T stage. How do the authors explain this finding?

Did the authors only look for lymph node recurrence? If so how was the lymph node recurrence confirmend? Was each patient confirmed by fine needle aspiration? CT diagnosis of lymph node metastases is notoriously difficult and contains significant errors. Data based just on this findings needs to be handled with caution.

Did the authors find any differences in 3 or 5 year survival? These would be hard criteria for the importance of IFITM3 analyses. Just CT lymph node findings are not convincing to the reviewer.

Additional comments

The submitted manuscript of Jia et al. analyses the role of IFITM3 expression in patients with pN0 esophageal squamous cell carcinoma after R0 resection. The findings could be of potential interest in the field to further stratify these patients. However a revision of the manuscript to clarify the findings is mandatory before publication.

---

## Round 0.2 · Minor Revisions

The manuscript is much improved. However, it still contains typographical errors. These need to be corrected prior to acceptance. Read the manuscript carefully for these and for other grammatical errors.

Reviewer 1 ·

Basic reporting

The manuscript appears to adhere to all PeerJ policies and is written with standard formatting and structure. I agree that parts of this manuscript are difficult to read and need to be proofread by a native English speaker.

Experimental design

The authors sought to determine whether IFITM3 serves as a predictor of regional recurrence in pN0 patients undergoing surgical treatment of ESCC. They analyzed 104 patients between 2008 to 2010 that were treated with Ivor-Lewis esophagectomy and two-field lymph node dissection. This study included only pN0 patients to identify predictors of regional recurrence. Patients did not receive adjuvant treatment.

IFITM3 expression was determined by IHC and RT-PCR in 20 specimens for confirmation.

Validity of the findings

Authors identified overexpression of IFITM3 in 56.7% of cases and that this expression was statistically related to T stage. 40.4% of patients developed regional recurrence within 3 years, and of these that recurred, 30/42 (71%) had IFITM3 overexpression, whereas in low IFITM3 expression the regional recurrence was 26.7%. The authors' statistical analysis showed that IFITM3 was thus a significant predictor of regional recurrence in this cohort. Multivariate analysis showed that IFITM3 was an independent predictor and not just a surrogate marker (such as for advanced T stage).

Additional comments

The authors have submitted revisions and a significantly stronger manuscript and addressed the question of recurrence rate. They have also provided more information on the surgical procedure and provided citations to previously published works. I do think this work is a valuable addition to the literature. I recommend that it be carefully proofread prior to final publication as several typographical errors are present.

·

Basic reporting

The authors answered the questions of the editor. However the authors omitted to answer sufficiently to the question concerning overall survival and claim that this will be the topic of a future article. To the reviewer this does not seem to make sense as it is just slicing an article into several less relevant ones. I would encourage the authors to include this data into the submitted manuscript. Otherwise the article can be now accepted.

Experimental design

The experimental design is of a sufficient standard.

Validity of the findings

The findings seem to be valid.

---

## Round 0.3 · accepted · Accept

It appears that the typographical and grammatical errors have been fixed.